# Brief communication: Intercomparison study reveals pathways for improving the representation of sea-ice biogeochemistry in models

Letizia Tedesco[1], Giulia Castellani[2,3], Pedro Duarte[4], Meibing Jin[5], Sebastien Moreau[4], Eric Mortenson[6], Benjamin Tobey Saenz[7], Nadja Steiner[8,9,10], Martin Vancoppenolle[11]

[1]Marine and Freshwater Solutions, Finnish Environment Institute, Helsinki, Finland
[2]Institute of Environmental Physics (IUP), University of Bremen, Bremen, Germany
[3]Deutsches Zentrum für Luft- und Raumfahrt (DLR), Institut für Physik der Atmosphäre, Oberpfaffenhofen, Germany
[4]Norwegian Polar Institute, Tromsø, Norway
[5]University of Alaska Fairbanks, AK, USA
[6]University of Miami, Cooperative Institute for Marine and Atmospheric Studies (CIMAS)
[7]Biota.Earth, Berkeley, CA, USA
[8]Institute of Ocean Sciences, Fisheries and Oceans Canada, Sidney, BC, Canada
[9]Canadian Center for Climate Modelling and Analysis, Environment and Climate Change Canada, Victoria, BC, Canada
[10]School of Earth and Ocean Sciences, University of Victoria, Victoria, BC, Canada
[11]Laboratoire d'Océanographie et du Climat, CNRS/IRD/MNHN, Sorbonne Université, Paris, France

*Correspondence to*: Letizia Tedesco (letizia.tedesco@environment.fi)

**Abstract.** Sea-ice biogeochemical models are key to understanding polar marine ecosystems. We present an intercomparison of six one-dimensional models, assessing their ability to simulate algal phenology and nutrient dynamics using physical-biogeochemical data from an Arctic drift expedition in spring 2015. While no model fully captured observed bloom dynamics with default settings, tuning improved biomass but had a limited impact on nutrients. The experiment revealed challenges in simulating short-lived, dynamic ice habitats, which are expected to become more common in a changing Arctic. Variability in tuning strategies underscores key knowledge gaps and highlights the need for coordinated future model developments to improve reliability and predictive capacity.

## 1 Introduction

Sea ice is home to an active microbial community, with ice algae displaying some of the highest Chlorophyll-a (Chl-a) concentrations of any aquatic environment (Arrigo, 2017). Ice algae play multiple pivotal roles in polar oceans, representing the largest biomass fraction in sea ice (Poulin et al., 2011), contributing to overall marine primary production (Dalman et al, 2025), acting as a critical food source for the marine food web, especially during winter (Schaafsma et al., 2017), and efficiently contributing to the ocean carbon sink (Boetius et al., 2013). Together with phytoplankton, ice algae form the foundation of the polar marine food web, supporting key under-ice foraging species such as Arctic cod (*Boreogadus saida*) in the Arctic Ocean (Geoffroy et al., 2023) and Antarctic krill (*Euphausia superba*) in the Southern Ocean (Kohlbach et al., 2017). These species

depend on the presence of sea ice and play a crucial role in transferring carbon to higher trophic levels, including humans (Steiner et al., 2021).

Current environmental changes are placing considerable pressure at the base of the food web, triggering significant effects throughout trophic levels (e.g., Post et al., 2013; Koch et al., 2023). Despite the recognised importance of the sea-ice ecosystems (Lannuzel et al., 2020), our knowledge remains limited due to their remote location and extreme weather conditions, which restrict observational data - particularly biological observations - to sparse spatial and temporal distributions. As a result, the representation of sea-ice biological and ecological processes in numerical models has historically been limited. However, in recent decades, significant advances have been made in modelling sea-ice habitats and the evolution of sea-ice biological communities (Castellani et al., 2025). Progress includes improved representation of physical processes, greater biodiversity, and enhanced ecosystem complexity.

An intercomparison of three-dimensional models has already been conducted to understand similarities and differences in simulated ice algae abundance and distribution, the Ice Algae Model Intercomparison Project – Phase 1 (IAMIP1, Watanabe et al., 2019). This study investigated the seasonal-to-decadal variability in ice-algal primary productivity across four Arctic regions during 1980–2009, as simulated by five participating models. Its conclusions indicated that, despite the ongoing reduction in Arctic sea ice, the decadal trend in ice-algal productivity remained unclear. The vernal bloom shifted towards an earlier onset and shorter duration over the simulated period, and the choice of maximum algal growth rate was identified as a key driver of inter-model differences in simulated ice-algal primary productivity. A second phase, expanding the study's scope to global coverage and centennial timescales following CMIP6 (Coupled Model Intercomparison Project Phase 6, Eyring et al., 2016) protocols, is currently underway (IAMIP2, Hayashida et al., 2021). However, given the numerous limitations and uncertainties associated with these large-scale models, they are more useful for deriving bulk properties than for investigating more detailed ecological processes.

To this end, one-dimensional (1D) process models become essential for addressing knowledge gaps in sea-ice biogeochemistry and ecological dynamics, as they provide a level of detail that large-scale models lack. They also allow for direct comparisons with in-situ observations, improving the ability to validate results. However, existing process models have been developed independently during periods of limited observations and incomplete process understanding, validated by observations at different locations, leading to substantial differences across models. These differences make an intercomparison of models performances challenging. To address this, the BEPSII (Biogeochemical Exchange Processes at Sea-Ice Interfaces, https://www.bepsii.org) expert group initiated an intercomparison of 1D sea-ice biogeochemical models, presented here, aiming at: i) understanding variability among models in representing key processes and responses to a common set of boundary conditions, ii) identifying divergences in models' behaviour, the variety of tuning strategy, and the drivers of model sensitivity, iii) testing transferability, and finally iv) promoting harmonisation for future model developments. The focus has been on

understanding the similarities and differences in simulated ice algae dynamics and investigating the controlling factors
responsible for the temporal variability and magnitude of ice-algal productivity among participating 1D models.

We present in this study an intercomparison of 1D sea-ice biogeochemical models (briefly described in Sect. 2.1 and more
comprehensively in Appendix A), focusing on their ability to simulate ice algal dynamics and nutrient cycling. Using a refrozen
lead time series (described in Sect 2.2) as a test case, we assess model performance through a structured comparison of
simulated and observed biogeochemical variables. Two experiments - *no tuning* and *tuning* - were conducted (Sect 2.3) to
evaluate the baseline model configurations as well as the impact of targeted parameter adjustments on model accuracy. We
analyse differences in model outputs, identify key sources of variability, and discuss the challenges associated with simulating
ice algal growth and nutrient fluxes (Sect 3). Finally, we highlight the implications of our findings for future model
development and propose directions for improving the representation of biogeochemical processes in sea-ice models (Sect. 4).
**2 Methods**
**2.1 Sea-ice biogeochemical models**
1D process models are typically designed to represent only vertical processes, assuming that horizontal advection is negligible.
Since they are computationally efficient, these models can incorporate a high level of ecosystem complexity, such as
representing multiple functional groups of organisms and providing high vertical resolution by discretising sea ice into several
layers.

1D sea-ice biogeochemical models vary in vertical resolution, ecosystem complexity, and whether they are coupled to the
ocean and/or atmosphere (Castellani et al., 2025). The biogeochemically active part of sea ice, also known as the Biologically
Active Layer (BAL) (Tedesco et al., 2010), is represented either as a single layer near the ice-ocean interface of prescribed or
variable thicknesses depending on sea-ice permeability, or as multiple layers spanning the vertical range of the sea ice with an
active brine network (e.g., Jeffery et al., 2016). Single-layer approaches are computationally more efficient than multi-layer
models. A single-layer model of variable thicknesses in response to thermodynamic growth, often referred to as dynamic
layering, provides a more realistic representation of bottom community dynamics (Tedesco et al., 2010). Multi-layer models,
on the other hand, capture the vertical variability of biogeochemical variables and allow simulating surface and infiltration
communities.

As in ocean models, the structure of sea-ice microbial ecosystems is represented using a set of "Plankton Functional Types"
(PFTs), which in our model framework include sea-ice algae, sea-ice heterotrophic bacteria, and sea-ice fauna such as grazers,
and non-living inorganic (e.g., sea-ice micro- and macronutrients) and organic matter (e.g., sea-ice detritus). The simplest
models are N-P models, which include only one nutrient (N) and one algal functional type (P). The elemental composition of

ice algae is typically fixed, based on prescribed Redfield carbon, nitrogen, silicon, phosphorous ratios (106:16:16:1), along with fixed Chl-a:carbon ratios. The more comprehensive N-P-Z-D models also include grazers (Z) (such as sea-ice fauna) and sea-ice detritus (D). In the simplest version of these models, only one limiting nutrient is considered. More complex models may represent multiple nutrients and different PFTs for ice algal communities, as well as bacteria and grazers. In simpler models, the processes associated with bacterial remineralisation or grazing are often implicitly parameterised using constant rates.

The intercomparison included five modelling teams and a total of six model configurations. These models varied in several aspects, encompassing differences in physical and biogeochemical process complexity, radiation schemes, vertical resolution, choice of limiting nutrient, area of original tuning of the model, and coupling to an interactive sea-ice physical model and/or ocean biogeochemical model of various complexity. Table 1 summarises the main commonalities and differences among the models. For more details on a specific model, we refer to the model's original reference (Table 1) and further description in Appendix A.

Most of the models had interactive physical components, while only one (i.e., SIMBA) required prescribed ice physics. Additionally, only half of the models were coupled to an interactive ocean biogeochemical model. Among the sea-ice physical models, complexities ranged from a Semtner 0-layer scheme (SM 0L) to multi-layer energy-conserving models (EC ML). All models, except one, used a single-band radiation transfer scheme, with several assuming Beer-Lambert (BL) light attenuation, while only one employed a Delta-Eddington (DE) scheme. The majority of the models simulated ice algae only in the bottom sea-ice layer, either as a static or dynamic system, while two models were multi-layer models, simulating ice algae along the entire ice column. In terms of ecosystem complexity, models varied from simple Redfield-based models (RFD) with a single limiting nutrient, one algal group, and a detritus compartment to more comprehensive quota models with several functional groups, including ice bacteria, ice fauna, and multi-nutrient limitations.

## 2.2 The N-ICE2015 Dataset

The refrozen lead time series monitored during the N-ICE2015 expedition (Granskog et al., 2018) was selected as a test case for the model intercomparison due to the high frequency of available physical and biogeochemical measurements (e.g. Kauko et al., 2017; Olsen et al., 2017). The N-ICE expedition was a field campaign conducted aboard the RV *Lance,* which was frozen into pack ice north of Svalbard, drifting between approximately 83° and 80°N in the southern Nansen Basin of the Arctic Ocean between January and June 2015. Among the four ice floes monitored during the study period, the refrozen lead data were derived from Floe 3, which was studied from mid-April to early June 2015 as it drifted southward from 81.8° N to 80.5° N.

**Table 1:** Sea-ice biogeochemical models participating in the 1D intercomparison project. BGC stands for biogeochemistry. Please see the main text for the remaining nomenclature used in the table.

| Model/ Properties | BFM-SI | BFM-SI-Clim | CICE 5.1 | CSIB-1D | SIESTA | SIMBA |
|---|---|---|---|---|---|---|
| **Ice Physics** | Modified SM 0L | Modified SM 0L | EC ML | SM 0L | EC ML | Prescribed |
| **Transport** | Growth/melt | Growth/melt | Growth/melt, brine drainage/diffusion | Melt | Desalination | Growth/melt |
| **Radiation** | 1 band; BL | 1 band; BL | 1 band; BL | 1 band; BL | 32 bands; DE | 1 band; BL |
| **Grid for sea ice BGC** | 1L, bottom, dynamic | 1L, bottom, dynamic | Multi-layer | 1L bottom static | Multi-layer | 1L bottom static |
| **Sea-ice functional groups** | 4N-2P-2D-1B-1Z | 1N-1P-2D | 3N-1P-1D | 3N-1P-1D | 4N-1P-1D | 1N-1P-1D |
| **Cell quotas/Chl:C** | Quota/Prognostic | Quota/Prognostic | RFD/Constant | RFD/Constant | RFD/Constant | RFD/Constant |
| **Limiting element(s)** | Nitrogen, Phosphorous, Silicon | Silicon | Nitrogen, Silicon | Nitrogen, Silicon | Nitrogen, Phosphorous, Silicon | Nitrogen |
| **Ocean BGC** | 1D slab | 1D slab | n.a. | 1D | n.a. | n.a. |
| **Area of model original tuning** | Greenland fjord (Arctic) | Greenland fjord (Arctic) | Barents Sea (Arctic) | Resolute Passage (Arctic) | Weddel Sea (Antarctic) | Central Arctic Ocean (Arctic) |
| **Reference** | Tedesco et al (2010) | Tedesco and Vichi (2014) | Duarte et al (2017) | Mortenson et al (2017) | Saenz and Arrigo (2014) | Castellani et al (2017) |

The lead, approximately 400 m wide, opened on 23 April, began refreezing on 26 April, and was fully refrozen by 1 May. The newly formed young ice in the lead was sampled from 6 May along a 100 m-long transect extending from the edge of the lead toward its centre every 2–3 days until it broke up on 4 June (Kauko et al., 2017). The algal growth period occurred in April and May. While the ice algal community was initially highly mixed, pennate diatoms of the genus *Nitzschia* became dominant later in the season.

The N-ICE2015 refrozen lead time series was chosen for this intercomparison based on two key factors:

- Observational data availability: It provides sufficient observations (Kauko et al., 2017) for comparison with model simulations of physico-biogeochemical variables.
- Ancillary data availability: It includes detailed time series of atmosphere and ocean data, necessary to force model runs, and has been tested for feasibility in a previous 1D modelling study (Duarte et al., 2017).

**2.3 Experimental setup**

A strict protocol was developed and followed by all modelling groups. To accommodate the diversity of models, a minimum set of variables was selected for comparison with observations. These included sea-ice season timing, ice thickness, and snow thickness for coupled physical-biogeochemical models, as well as sea-ice nutrient concentrations and algal biomass (represented by Chl-a) for all models.

Two distinct experiments were conducted to assess model performance. The first experiment, labelled *no tuning*, aimed to run each model in its default configuration. The primary objective was to analyse the differences between model outputs and observational data and quantify the extent of biases. The intercomparison within this experiment sought to identify potential reasons for deviations from observations, such as the omission of key processes or inadequate parameterisations. The second experiment, labelled *tuning*, involved adjusting the models to better align with observed physical and biogeochemical properties. This experiment aimed to identify which processes needed to be modified or added, as well as the specific parameterisations or parameters that were adjusted and fine-tuned to improve agreement with observations.

Both experiments were carried out independently by each modelling group, without prior knowledge of the work undertaken by others. This approach was adopted to eliminate potential biases, whether conscious or unconscious, during the implementation phase. To ensure a standardized comparison across models, all simulations used the same atmospheric and ocean forcing, as well as identical initial and boundary conditions, described in Duarte et al. (2017). Forcing time series included air temperature, precipitation, specific humidity, and wind speed (Hudson et al., 2015; Cohen et al., 2017); incident surface short and longwave radiation (Taskjelle et al., 2016; Hudson et al., 2016); sea ice temperature and salinity (Gerland et al., 2017); surface current velocity, heat fluxes, salinity, and temperature (Peterson et al., 2016, 2017); and ocean surface nutrient concentrations (Assmy et al., 2016). Atmospheric forcing was provided at hourly resolution, while oceanic forcing was available daily. For the sea-ice biogeochemical model without a thermodynamic component (i.e., SIMBA), observed ice and snow thickness data were provided. This standardised approach improved the comparability of the models, allowing for a robust evaluation of model performance. In the final phase, results were presented by each modelling group, and teams collaboratively discussed challenges, adjustments, and tuning choices.

## 3. Results and discussion

To support the interpretation of the biogeochemical models' performances, we first compared modelled and observed sea-ice physical properties, in particular sea-ice thickness and surface (snow/ice) temperature (Fig. 1). While the models were forced with 2 m air temperature, the surface temperature shown here refers to the simulated snow or ice surface temperature, which may diverge from the atmospheric forcing depending on the model physics and surface energy budget. We did not include snow thickness in this comparison, as observed values were relatively low and little variable, ranging between 2 and 6 cm between 7 May and 3 June (Kauko et al., 2017) and thus had a limited influence on model differences for this specific case.

Observed sea-ice thickness shows relatively stable values around 0.2 m from early May to early June, with minor variability in the observations (Fig.1). Models with thermodynamic components (BFMSI/BFMSI-CLIM, CICE5.1, CSIB-1D, and SIESTA) generally captured the observed thickness range and seasonal trend, although some diverge more notably. Surface temperature simulations show stronger deviations across models. Although all models follow the overall seasonal warming trend observed in the N-ICE2015 air temperature data (Fig. 1, right panel), the amplitude and short-term variability differ. While some models reproduce much of the daily variability, others exhibit smoother or warmer biases. These differences in physical conditions influenced light penetration and melt timing, which in turn affected the timing and magnitude of simulated algal blooms, which will be analysed next.

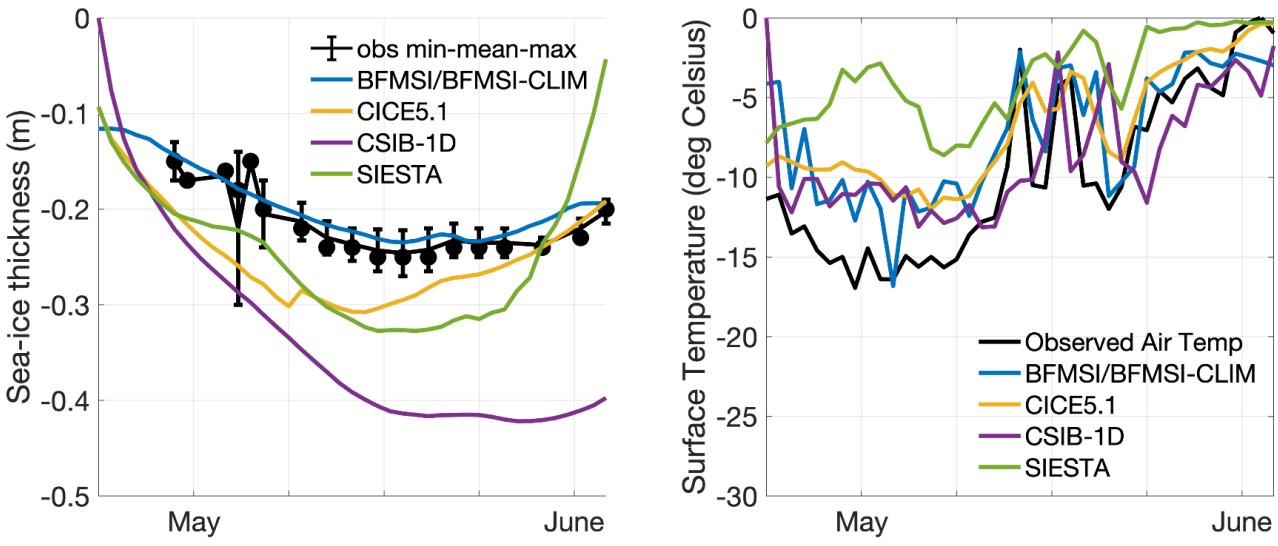

**Figure 1:** Model results for sea-ice thickness (left) and surface temperature (right). Observations of sea-ice thickness are shown as dots for the mean among replicates (at least 5 each) from different ice cores, while associated bars indicate the variability of the measurements between their maximum and minimum. The observed air temperature is part of the forcings provided to the modelling groups (Hudson et al., 2015; Cohen et al., 2017) and it is shown for comparison with modelled surface temperature.

Although the N-ICE refrozen lead resembles a typical ice season, in the *no tuning* experiment, none of the models accurately captured the observed algal phenology and bloom magnitude (Fig. 2, top left). All but one model underestimated Chl-a and produced a delayed bloom onset, though performances varied across diagnostic measures. Since most of the models tended to overestimate sea-ice thickness (Fig. 1), the delay in the simulated algal bloom could be attributed to reduced light transmittance through thicker ice. However, the delay also occurred in models that did not overestimate ice thickness, suggesting that other factors must had contributed to this bias. Due to limited nutrient data, few considerations can be drawn about simulated nutrient dynamics beyond an assessment of the potential model error's order of magnitude. Here, all but one model underestimated nitrate and silicate concentrations (Fig. 2, mid and bottom left), though all remained within a reasonable range.

In the *tuning* experiment, all models were able to reasonably simulate the ice algal phenology, though performance still varied across models (Fig. 2, top right). However, little improvement was achieved in the simulation of nitrate and silicate dynamics. Interestingly, tuning focused on different processes and parameters among models (Table 2), including:

- Change in the algal growth rate and/or in the size of the initial seeding population (initial ice algal biomass)
- The possibility of downward vertical migration of algae during melting
- Magnitude of silicic acid limitation by changing the half saturation constant and/or the nitrogen: silicon ratio of ice algae and/or the reference quota of silicon in sea-ice algae.

Overall, all tuning strategies aimed to either lessen nutrient limitation or increase algal seeding or growth. However, despite tuning efforts, none of the models significantly improved the simulation of nitrate magnitude, except for BFM-SI, which was also the only model that did not underestimate nitrate and silicate before tuning (Fig. 2, mid and bottom left). When comparing nutrient parameterisations across models (Table 1), BFM-SI stands out as the only model in which the variability of the dynamic sea-ice BAL modulates the upward fluxes of dissolved inorganic matter. CSIB-1D also performed well in simulating the silicate dynamics, matching the magnitude of the observations before and after tuning. For most models, silicon had the strongest effect on ice algal growth during tuning, suggesting a potentially dominant role of silicon limitation. This would also explain why SIMBA was the only model that did not underestimate, but rather overestimated, ice algal growth, since it did not include silicon among its limiting nutrients.

In general, models performed more poorly when simulating sea-ice nutrient dynamics. The limited improvement in nutrient representation compared to biomass can be attributed to model groups prioritising fitting simulations to Chl-a observations during the tuning phase, as these data were more temporally resolved and directly linked to the main focus of the study, i.e., the ice algal bloom. In contrast, nutrient observations were limited to a single time point, which made them more difficult to constrain reliably. Nevertheless, despite the scarcity of available data, the simulation of nutrient processes appears poorly constrained, pointing to the need for more in-depth observational and experimental work.

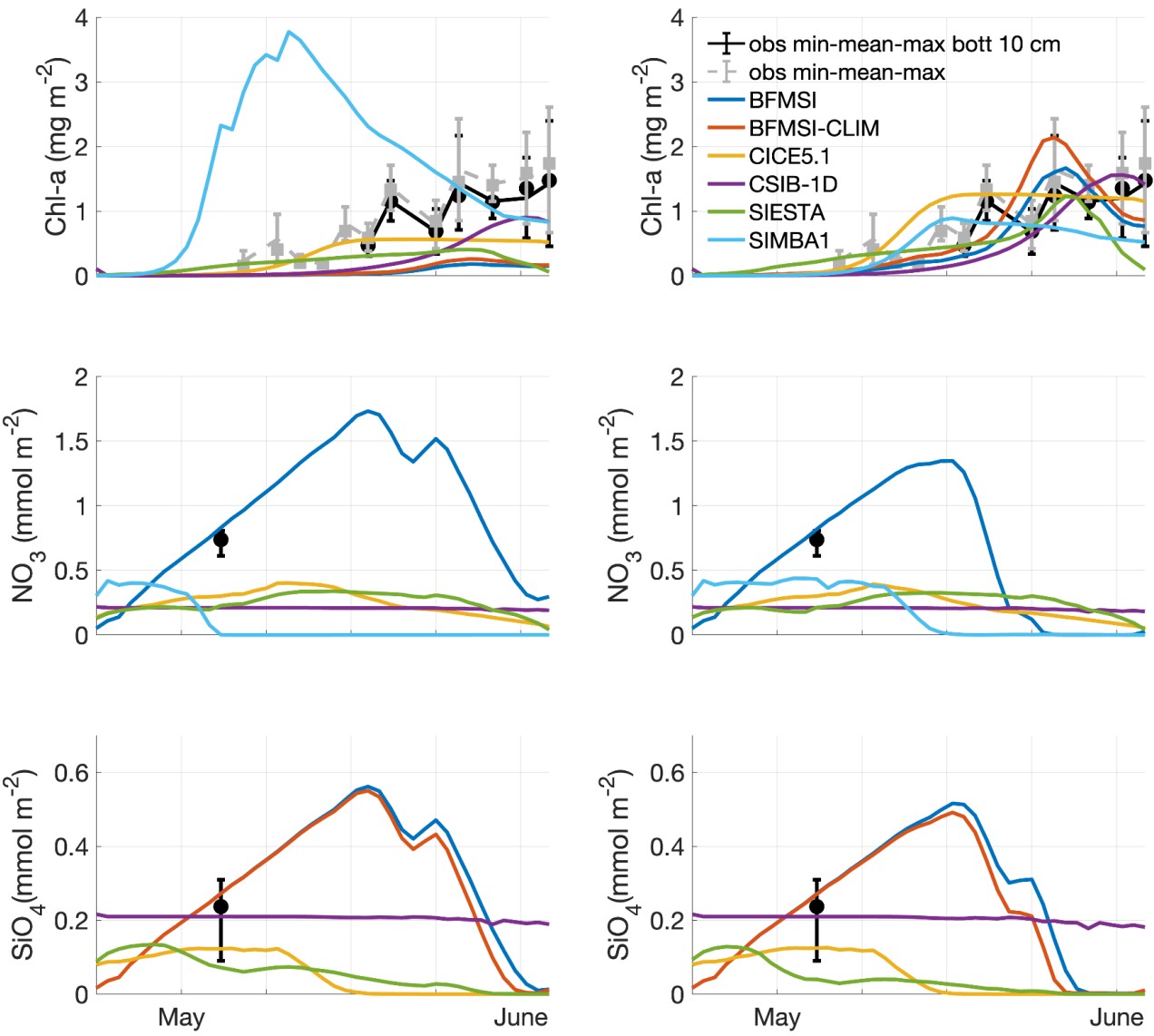

229

**Figure 2:** Experiment with *no tuning* (left) and *tuning* (right). Model results for ice algae Chl-a (top), nitrate (middle), and silicate (bottom). Observations are shown as dots for the mean of the entire ice core or the bottom 10 cm (5 replicates each), while associated bars indicate the variability of the measurements between their maximum and minimum measures.

The *tuning* experiment highlights the diversity of tuning parameters across models (Table 2), prompting critical questions about model functionality and calibration. While models can be adjusted to align with observations, there is a risk of achieving

accurate results for the wrong reasons, particularly when tuning compensates for a missing or misrepresented process. In our
case, none of our models included young ice formation. Observations indicate that a consistent fraction of the sea-ice sampled
from the refrozen lead was granular (Graham et al., 2019), formed as frazil ice in turbulent conditions. As turbulence subsides,
frazil crystals rise and can entrain suspended particles, including biological material, during acent, effectively concentrating
them in the newly forming ice (Weeks and Ackley, 1982, Janssen et al., 2018). This may explain some of the tuning strategies,
such as increases in algal growth rate (CSIB-1D) or the size of the initial seeding population (BFM-SI, BFM-SI-Clim).

However, other factors likely influenced tuning choices as well. For example, some models used diatom Si:N ratios more
appropriate for Antarctic waters, which overestimate the silica demand of Arctic diatoms. For example, CICE used a Si:N ratio
close to 4:1, whereas Arctic diatoms may be closer to 1:1 (Duarte et al., 2017). In addition, the presence of relatively low Si:N
ratios in Atlantic Water entering the region, as discussed in studies such as Duarte et al. (2021), supports the potential for silica
limitation to emerge before nitrogen is exhausted. These regional nutrient characteristics and model structural features may
have prompted tuning strategies involving relaxed silica limitation (BFM-SI, BFM-SI-Clim, CICE 5.1, and SIESTA).
Furthermore, the apparent need to reduce nutrient limitation in order to simulate realistic biomass may indicate that ocean-to-
ice nutrient fluxes are underestimated in some models (Duarte et al., 2022).

Taken together, this intercomparison underscores how model tuning decisions can reveal not only numerical sensitivities but
also areas where physical and biogeochemical process representations remain uncertain or incomplete. These insights are
valuable for guiding future model development and targeted observations.

## 4. Conclusions

This study presents an intercomparison of one-dimensional sea-ice biogeochemical models, evaluating their ability to simulate
algal phenology, bloom magnitude, and nutrient dynamics in a refrozen lead environment. The results highlight significant
disparities in model performance, with most models struggling to accurately reproduce the observed algal biomass and nutrient
concentrations. For some models, this difficulty persisted even after tuning. While adjustments improved the representation of
ice algal phenology, they had a limited impact on nutrient concentration across most models, emphasizing the challenges of
parameterizing key processes such as nutrient fluxes and reinforcing the need for continued model development and validation
supported by dedicated field and experimental observations.




**Table 2.** Comparison among models' performances before and after tuning. For reference, observed chlorophyll-a concentrations peaked
on 3 June at ~2.1 mg m$^{-3}$ in bottom sea ice and ~2.6 mg m$^{-3}$ in whole sea ice, aiding comparison of simulated bloom timing and magnitude.
Only parameters that were explicitly tuned are listed. Parameters not shown were kept at their default values or followed the standard initial
and boundary conditions provided for the intercomparison.

| Model/ Properties | BFM-SI | BFM-SI-Clim | CICE 5.1 | CSIB-1D | SIESTA | SIMBA |
|---|---|---|---|---|---|---|
| **Ice algal phenology before tuning** | Good algal growth timing but lower algal biomass. <br><br> Max [Chl-a] = 0.18 mg m$^{-2}$ <br><br> Day of the year of peak of Chl-a = 146 | Good algal growth timing but lower algal biomass. <br><br> Max [Chl-a] = 0.26 mg m$^{-2}$ <br><br> Day of the year of peak of Chl-a = 146 | Good algal growth timing but lower algal biomass. <br><br> Max [Chl-a] = 0.56 mg m$^{-2}$ <br><br> Day of the year of peak of Chl-a = 142 | Good algal growth and lower algal biomass. <br><br> Max [Chl-a] = 0.90 mg m$^{-2}$ <br><br> Day of the year of peak of Chl-a = 152 | Good algal growth timing but lower algal biomass. <br><br> Max [Chl-a] = 0.41 mg m$^{-2}$ <br><br> Day of the year of peak of Chl-a = 147 | Earlier algal growth and higher algal biomass <br><br> Max [Chl-a] = 3.77 mg m$^{-2}$ <br><br> Day of the year of peak of Chl-a = 131 |
| **Tuning strategy** | Lower silica limitation and higher algal biomass in seawater | Lower silica limitation and higher algal biomass in seawater | Lower silica limitation and reduced recruitment | Higher algal max spec growth rate | Active algal migration against brine movement and lower Si half-saturation constant. | Lower algal growth rate and removal of winter drainage of nutrients |
| **Parameter(s) before tuning** | Initial seawater [Chl-a] =0.05 mg m$^{-3}$ <br><br> Reference Si quotum for adapted diatoms=0.0085 mmol m$^{-3}$ | Initial seawater [Chl-a] =0.05 mg m$^{-3}$ <br><br> Reference Si quotum for adapted diatoms=0.0085 mmol m$^{-3}$ | Diatom Si:N ratio = 1.8 Half saturation for silicon uptake = 4.0 μM Diatom boundary concentration = 0.002 μM | Chl-a max spec growth rate = 0.85 d$^{-1}$ | Algae fixed in ice layer grid; Half saturation of silicon uptake = 4.0 μM | Chl-a max spec growth rate = 0.86 d-1 |
| **Parameter(s) after tuning** | Initial seawater [Chl-a] in =0.5 mg m$^{-3}$ <br><br> Reference Si quotum for adapted diatoms=0.0025 mmol m$^{-3}$ | Initial seawater [Chl-a] in =0.5 mg m$^{-3}$ <br><br> Reference Si quotum for adapted diatoms=0.0025 mmol m$^{-3}$ | Diatom Si:N ratio = 1.0 Half saturation for silicon uptake = 2.2 μM Diatom boundary concentration = 0.0011 μM | Chl-a max spec growth rate increased to 0.95 d$^{-1}$ | Algae allowed to migrate downward with ice growth, up to 1.5 cm d$^{-1}$; Half saturation of silicon uptake = 1.0 μM | Chl-a max spec growth rate = 0.5 d-1 |

| Ice algal phenology after tuning | Algal phenology and magnitude within observed range; Nitrate and silicate within range. | Algal phenology and magnitude within observed range, Silicate within range. | Algal phenology and magnitude within observed range; Lower nitrate, Silicate within range. | Algal phenology and magnitude within observed range; Lower nitrate; Silicate within range. | Algal phenology within observed range; Earlier algal decay; Lower silicate and nitrate. | Algal phenology and magnitude within observed range; Lower nitrate. |
|---|---|---|---|---|---|---|
| | Max [Chl-a] = 1.67 mg m$^{-2}$ | Max [Chl-a] = 2.14 mg m$^{-2}$ | Max [Chl-a] = 1.26 mg m$^{-2}$ | Max [Chl-a] = 1.56 mg m$^{-2}$ | Max [Chl-a] = 1.23 mg m$^{-2}$ | Max [Chl-a] = 0.89 mg m$^{-2}$ |
| | Day of the year of peak of Chl-a = 146 | Day of the year of peak of Chl-a = 147 | Day of the year of peak of Chl-a = 141 | Day of the year of peak of Chl-a = 152 | Day of the year of peak of Chl-a = 147 | Day of the year of peak of Chl-a = 137 |

The intercomparison highlights the unexpected challenges encountered in simulating a refrozen lead, primarily attributed to the short ice season and the difficulty most models faced in accumulating sufficient sympagic (i.e., in-ice) biomass. In a future Arctic Ocean characterized by increased lead openings, refreezing events, and young ice formation, there is an urgent need for models to be able to represent such a dynamic environment. This study underscores the importance of understanding and addressing the complexities involved in simulating specific and dynamic environmental scenarios.

The diversity of adjustments across models highlights both the range of tuning options available and the persisting knowledge gaps. The insights gained contribute valuable knowledge to ongoing efforts aimed at refining and improving numerical models, ensuring their accuracy and reliability in capturing complex interactions. To further advance this field of science, collaborative and harmonized modelling developments are recommended. Variability in tuning strategies underscores key knowledge gaps and the need for further model development using more coordinated approaches, such as common evaluation criteria and/or shared parameter ranges. In doing so, sea-ice biogeochemical modelling can build on lessons learned from open-ocean biogeochemical intercomparison and tuning efforts (e.g., Schartau et al., 2017), while addressing the unique challenges of simulating sympagic systems. A *Phase 2* of the intercomparison would be highly valuable, potentially extending the study to the variability of habitats that characterizes Antarctic sea ice. Collaborative sensitivity tests could be conducted, with all models evaluating biological responses to the same tuning adjustments, tuning options could be expanded, and standard parameter ranges could be revisited based on newer data collected in recent years. Increased clarity of model sensitivities would improve future model robustness and enhance confidence in simulations of biogeochemical processes in ice-covered oceans.

**Code and data availability**

All relevant data, model code and numerical simulations presented in this work will be publicly made available upon manuscript's acceptance.

## Author contributions

LT and MV conceived the study. LT, GC, PD, EM, and BS produced the model runs. LT merged results from different models and wrote the first draft of the ms. All authors contributed to the analysis of results, discussion, and/or editing of the manuscript.

## Competing interests

The authors declare no competing interests.

## Acknowledgements

This work was written under the auspices of BEPSII (Biogeochemical Exchange Processes at the Sea-Ice Interfaces) Expert Group (www.bepsii.org).

## Financial support

LT and PD received funding from the European Union's Horizon 2020 research and innovation programme under grant agreement No 101003826 via project CRiceS (Climate relevant interactions and feedbacks: the key role of sea ice and snow in the polar and global climate system). GC, PD, and SM received funding from the RCN BREATHE project (Bottom sea ice Respiration and nutrient Exchanges Assessed for THE Arctic, #325405). PD and SM also received funding from the iC3 Center of Excellence (Centre for ice, Cryosphere, Carbon and Climate, #332635). EM received funding through ArcticNet and the National Science and Engineering Council in Canada. NS acknowledges Fisheries and Oceans Canada. Two in-person workshops were organized by the BEPSII Expert Group (Biogeochemical Exchange Processes at Sea Ice Interfaces) to support this work. These workshops were financially supported by the Scientific Committee on Antarctic Research (SCAR) and the World Climate Research Programme (WCRP) Climate and Cryosphere (CliC) core project.





**Appendix A**

**A1 Models description**

*BFM-SI and BFM-SI-Clim*

*Overview*

The Biogeochemical Flux Model for sea ice (BFM-SI, Tedesco et al., 2010) is derived from the Biogeochemical Flux Model (BFM) framework (Vichi et al., 2023 and references therein), retaining its structure based on Chemical Functional Families (CFFs) and Living Functional Groups (LFGs). CFFs represent the elemental composition of living and non-living matter (C, N, P, Si, etc.), while LFGs describe groups of organisms with similar functional behaviour.

The model simulates biogeochemical processes within the Biologically Active Layer (BAL, Tedesco et al., 2010), the time-varying, permeable fraction of sea ice where liquid brine channels remain interconnected and biological activity can occur. This dynamic layer, typically located at the ice bottom, evolves according to physical conditions (e.g., temperature, salinity, brine volume) computed by a sea-ice physical model. The biological model simulates algal growth and elemental cycling only within this layer, assuming all biomass is confined to the permeable ice fraction continuously connected to seawater, maintaining full mass conservation at the ice–ocean–atmosphere interfaces.

The sea-ice physical model used in this study is ESIM (Enhanced Sea Ice Model). ESIM is a sea-ice thermodynamic model originally based on the Semtner 0-layer model (Semtner, 1976), but with more physical processes. It was initially built as a 1-D thermodynamic model of the sea-ice growth and decay (Tedesco et al., 2009), calculating vertical heat fluxes based on the 1-dimensional heat conduction equation. ESIM has been later enhanced with a halodynamic component (Tedesco et al., 2010). Initial salt entrapment, gravity drainage, and flushing processes have been added to simulate the salinity evolution of the sea ice. In addition, the model takes into account other processes such as different forms of snow metamorphism (snow compaction, snow ice and superimposed ice formation). ESIM has been developed targeting biological applications, thus with

a focus on the physical requirements to model the biogeochemistry of the sea ice. The feature that makes this coupling possible
is the innovative concept of the sea-ice BAL (Tedesco et al., 2010). The application of the BAL concept is more realistic than
a prescribed static bottom BAL and is lighter than multi-layer models, thus it is suitable for large-scale applications without
losing performance (Tedesco and Vichi, 2010, 2014).
*State variables and structure*
BFM-SI resolves 28 state variables organized as:
- 2 LFGs for sea-ice algae:
1. Adapted diatoms (20–200 µm; Si-limited, highly acclimated)
2. Surviving nanoflagellates (2–20 µm; low acclimation capacity)
- 1 LFG for sea-ice fauna
- 1 LFG for sea-ice bacteria
- 6 inorganic CFFs: phosphate, nitrate, ammonium, silicate, oxygen, carbon dioxide.
- 2 organic non-living CFFs: dissolved and particulate detritus.
Each algal group is described by up to five state variables (C, N, P, Si, and Chl), while ice fauna and bacteria up to three state
variables (C, N, P). The model includes four macronutrients (phosphate, nitrate, ammonium, silicate), oxygen, and two detrital
pools (dissolved and particulate, featuring up to 4 state variables C, N, P, Si). Biological processes include primary production
respiration, exudation, nutrient uptake, lysis, and chlorophyll synthesis, with flexible stoichiometry (C:N:P:Si:Chl).
BFM-SI-Clim (Tedesco et al., 2014) is a simplified version of BFM-SI, retaining the same ecological dynamics, but including
a reduced number of state variables. BFM-SI-Clim features only one single limiting macronutrient (Si) and one single group
of sea ice algae (i.e. ice diatoms), same detritus and gases for totally 11 state variables.
*Coupling and boundary fluxes*
BFM-SI and BFM-SI-Clim are coupled online to the pelagic BFM with matching LFGs and CFFs.
- Ice–ocean fluxes: The entrainment or release of dissolved and particulate matter is proportional to ice growth/melt
rate and brine volume.
- Ice–atmosphere fluxes: The nutrient input from snow and precipitation can be considered and scaled to snow-melt
rate.
These exchanges ensure conservation of mass and consistent carbon, nutrient, and gas cycling across the interfaces.

*Applications and relevance*

BFM-SI represents the first process-based, biomass-explicit sea-ice biogeochemical model within a generalized marine biogeochemical framework. It can be used as a standalone 1-D module (Tedesco et al., 2010; Tedesco et al., 2012; Tedesco et al., 2014) or in coupled online or offline configuration to 3-D ocean circulation models (Tedesco et al., 2017; Tedesco et al., 2019) to study seasonal productivity, biomass export, and the contribution of sea-ice biogeochemistry to the global carbon cycle.

**CICE 5.1**

*Overview*

A comprehensive description of the Los Alamos Sea Ice Model physics and biogeochemistry may be found in Hunke et al. (2015) and Jeffery et al. (2016). The implementation used in the present work is detailed in Duarte et al. (2017). Therefore, in the next paragraphs we provide only a brief description of the model based on the cited references. There are two main approaches to simulate biogeochemical processes with CICE: one based on bottom ice biogeochemistry and another based on vertically-resolved biogeochemistry, which was used in the present study. This configuration uses a biogrid of variable height which overlaps part of the physical grid, used to compute thermodynamic processes. The number of layers of both grids is the same but their vertical resolution differs. The vertical extent of the biogrid is defined by the brine height which represents the sea ice vertical extent with an active brine network.

*State variables and structure*

The number of biogeochemical state variables in CICE biogeochemistry depends on user-defined options. In the simulations presented herein, these included brine height, the concentrations of nitrate, ammonium, silicic acid and diatom nitrogen. Brine concentrations are used for internal calculations and bulk values stored in model output files. The brine is exchanged across the layers of the biogrid and across the ice-ocean interface. These exchanges include brine drainage, driven by hydrostatic instability, and diffusion, driven by concentration gradients. Other exchanges occur during freezing and melting. In the case of sea ice inundation or snow melt, exchanges occur also at the ice-snow or ice-atmosphere interface. The biogeochemical model uses nitrogen as its "currency". The model computes nutrient and silicic acid (in the case of diatoms) uptake by ice algae, remineralization and nitrification. Ice algal growth and production may be light, temperature or nutrient limited (nitrogen and silica, in the case of diatoms), following the Liebig's law of minimum. Some tracers may cling to the ice matrix, such as ice algae, resisting expulsion during desalination, unlike dissolved nutrients.

*Coupling and boundary fluxes*

The CICE model may be coupled with ocean models and atmospheric models. We used a standalone configuration with an
ocean slab layer as the bottom boundary. Time series of current velocities, heat fluxes, salinity, temperature, and nutrient
concentrations forced the model. The atmosphere boundary was implemented using time series of air temperature, humidity,
short and long wave radiation, precipitation, and wind velocity.
*Applications and relevance*
The CICE model is a community-type model used in several Earth System Models. It is one of the few models resolving
biogeochemistry vertically.

**CSIB-1D**
*Overview*
The Canadian Sea Ice Biogeochemistry 1-Dimensional (CSIB-1D) model simulates ice algae and changes to nutrients within
the ice. It is designed to simulate a sympagic ecosystem and biogeochemical processes coupled to a pelagic ecosystem in the
underlying water column in order to represent the Arctic marine environment. An in-depth description of the development and
application of this model can be found in Mortenson et al. (2017).
*State variables and structure*
The CSIB-1D ecosystem is represented by one functional sea-ice algal group dependent on three nutrients (silicate, nitrate and
ammonium) in the lower skeletal layer of the sea ice, set as a default in the bottom 3 centimetres of the ice. The sea ice algae
are limited by nutrients, light, and ice melt. The model uses a subgrid-scale non-uniform snow depth distribution to represent
gradual snow melt and formation of melt ponds impacting light transmissions and heat fluxes during melt periods (Abraham
et al., 2015). CSIB-1D ice algae are meant to represent diatoms, prevalent in the Arctic sea ice environment.
The ocean biogeochemistry model is a ten-compartment (small and large phytoplankton, microzooplankton, mesozooplankton,
small and large detritus, biogenic silica, nitrate, ammonium, and silicate) based on Steiner et al. (2006). The module was
updated by including mesozooplankton as a prognostic.
*Coupling and boundary fluxes*
Exchange of nutrients between the skeletal layer and the water column is by molecular diffusion and parameterized based on
currents at the ice-water interface. The model is coupled to a physical-biogeochemical ocean model based on the General
Ocean Turbulence Model (GOTM). GOTM provides the physical quantities required for computation of biogeochemical
variables in the water column, such as horizontal velocity fields, turbulent transports, photosynthetically active radiation
(PAR), and temperature. They contribute to pelagic diatoms and detritus following Lavoie et al. (2009): sloughed ice algae
enter either the large phytoplankton pool in which they continue to grow or the large detritus pool in which they sink rapidly
as aggregate products in the coupled ocean model.
*Application and Relevance*
CSIB has been applied to studies on the evolution of the ice-water exchange of dissolved inorganic carbon (Mortenson et al.,
2018) and ice-water-air exchange of dimethyl sulfide (Hayashida et al., 2017) in the marine Arctic.

**SIESTA**
*Overview*
The Sea-Ice Ecosystem State (SIESTA) model is a thermodynamic vertically-layered sea ice and snow model coupled to an
algal ecosystem model. The model and associated equations and parameterizations are described in Saenz and Arrigo (2012,
2014). The model was developed to vertically resolve sea ice brine processes (and associated nutrient transfer), sea ice optics,
shortwave radiation transfer, and the sea ice algal productivity that is controlled by those processes. The model uses a minimum
layer thickness of 2 cm. When the snow or ice thicknesses become greater than is resolved by the maximum number of layers
(snow: 26, ice: 42), model layers grow and shrink in an accordion-fashion to preserve 2 cm resolution at the surface and snow-
ice boundaries.
*State variables and structure*
Sea ice algae in SIESTA is represented by a single (diatom) class of algae with a fixed stoichiometry, with internal units of
carbon ($mg/m^3$). Algae may be present in any layer of sea ice. Besides algal carbon, the ecological state variables used by the
SIESTA model include temperature, salinity, density, particulate organic carbon (detritus that is remineralized to liberate
macronutrients), and 4 macronutrients (ammonium, nitrate, phosphate, silica). The model dynamically calculates sea ice brine
density and volume, and has parameterizations of snow metamorphosis, sea ice surface melt and ponding, snow-ice formation,
brine pumping and drainage, and enhanced convection in the skeletal layer of growing sea ice.  Sea ice algae are considered
motile and can migrate downward at a limited rate, but do not migrate upward and are considered released to the water column
during bottom ice melt.
*Coupling and boundary fluxes*
SIESTA simulations in this manuscript were forced by time series of surface atmospheric and surface ocean parameters.
SIESTA is mass- and energy-conservative to the accuracy of its 1st-order implicit solver. Coupling at the surface boundary
requires the following atmospheric parameters: air temperature, wind speed, air pressure, dew point temperature, cloud cover
(or downward longwave radiation), downwelling shortwave radiation) total precipitation.  Coupling at the lower boundary
requires the following surface ocean parameters: temperature, salinity, and macronutrient concentrations (ammonium, nitrate,
phosphate, silica). SIESTA calculates, and can return to coupled models, energy and mass fluxes from the snow/ice/brine.
Boundary flux calculations in SIESTA are derived from CICE version 4 (Hunke and Lipscomb, 2008).
*Applications and relevance*
SIESTA has been used to help bound the contribution of sea ice algae to overall Southern Ocean primary production (Saenz
and Arrigo, 2014).  SIESTA is also coupled to a 1-dimensional vertical ocean model (KPP-Ecosystem-Ice [KEI]) for
investigation of dynamic-thermodynamic sea-ice-ocean-ecosystem controls and interactions (Saenz et al. 2023).

***SIMBA***
*Overview*
A comprehensive description of the Sea Ice Model for Bottom Algae (SIMBA) can be found in Castellani et al. (2017).
Different from Castellani et al. (2017) where the process of growth/melt was responsible for only algal loss, in the present
study it is applied to nutrients as well, and it is responsible for nutrient replenishment in the bottom of the ice.
*State variables and structure*
SIMBA resolves only 3 state variables:
● 1 for sea-ice algae:
● 1 for nutrients (nitrate)
● 1 for detritus
The simulated biological processes are primary production and nutrient uptake, whereas respiration, mortality, and
remineralization are taken as constant. Equations are solved in mmol N m-3. Equations are solved in the bottom of the ice, the
thickness of the ice bottom can be set according to the available observations. In the case of N-ICE we use 10 cm.
*Coupling and boundary fluxes*
SIMBA is coupled with the underlying ocean through the growth and melt processes which are responsible for nutrient
exchanges and for algal loss. Ocean variables (i.e., nutrients concentrations, ocean currents, and ocean temperature) must be
provided as forcing. Other required forcing includes ice and snow thickness, integrated downward shortwave radiation, and
atmospheric temperature.
*Applications and relevance*
SIMBA was developed to study algal phenology on a pan-Arctic scale in two different environments: level ice and deformed
ice. With this aim, SIMBA requires a prescribed physics. In Castellani et al. (2017) the physical constraints were provided by
the MITgcm (Marshall et al., 1997; Losch et al, 2010). This characteristic of the model enhances its flexibility in applications
and studies with different models (see e.g., Castellani et al., 2021).

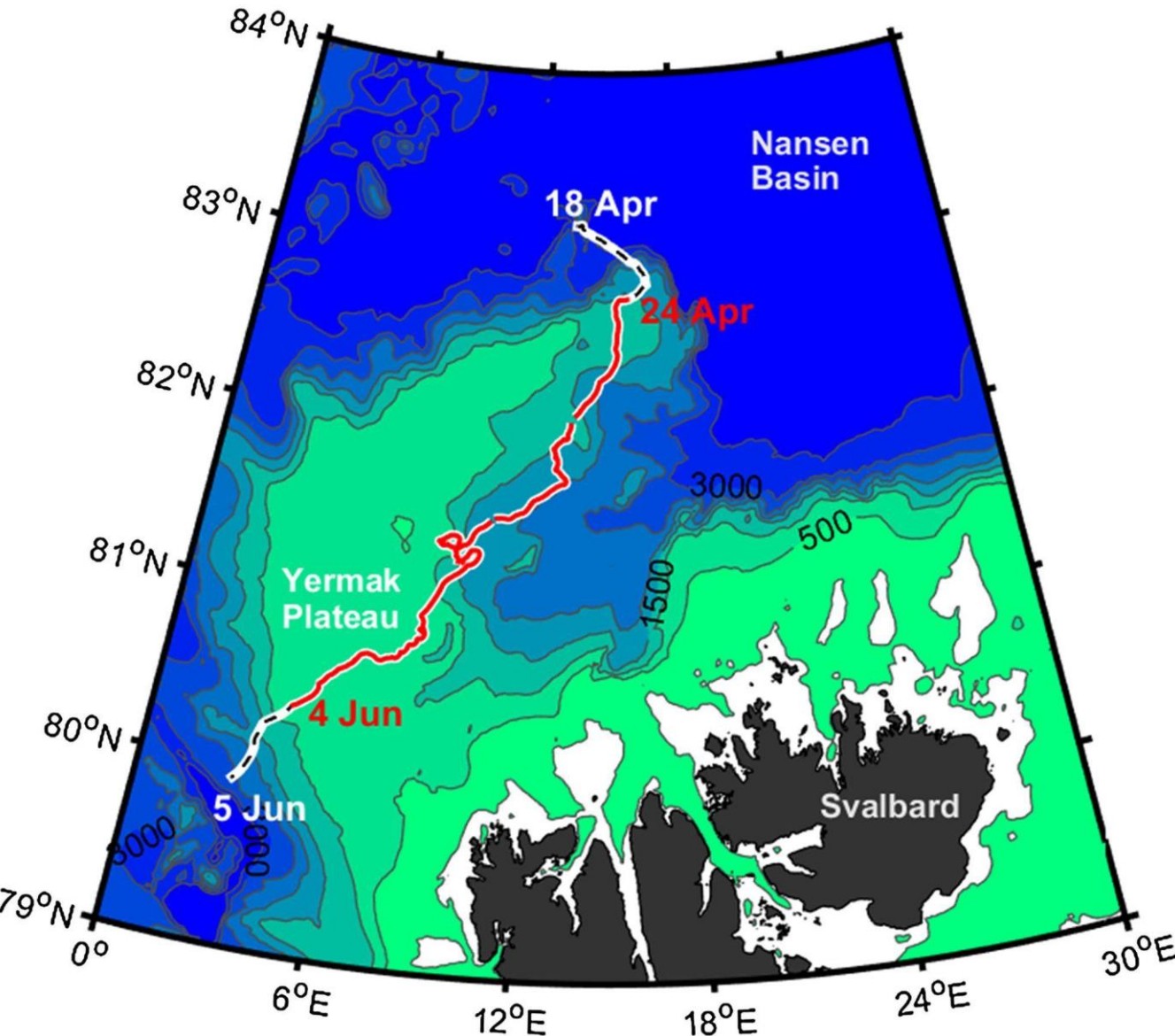

**Figure A1.** RV Lance drift between 18 April and 5 June 2015 during the drift of Floe 3 of the N-ICE2015 expedition, from the Nansen Basin and across the Yermak Plateau. The segment corresponding to the time span of the simulations described in this study is shown in red (Duarte et al., 2017).

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
