# Peer review of "Brief communication: Intercomparison study reveals pathways for"

_EGUsphere, 2025_

## Author Comment (AC1)

**Rebuttal "Brief communication: Intercomparison study reveals pathways for improving the representation of sea-ice biogeochemistry in models" by Tedesco et al., 2025**

Reviewer #1

The manuscript presents an intercomparison between 6 one-dimensional sea-ice biogeochemical models. Models were forced with the same atmospheric and oceanic data from a time series of observations from a refrozen lead and validated with associated chlorophyll *a*, nitrate, and silicate data. Models were first run with their default parameterizations, and then teams were asked to tune their models to best match the observations. The authors found that with tuning, most models were able to reproduce chlorophyll *a* timing and concentration but not so much nutrients. Model teams also differed in their tuning choices.

I think the exercise of model intercomparison is valuable and this is a timely moment to do one for sea-ice BGC, as the number of models has been increasing. I also think the exercise is well-structured and shows significant effort on the part of many people in independently running and tuning the models with the same set of forcings (not a given in other model intercomparisons) in order to reveal a process understanding.

We thank Reviewer #1 for their thoughtful and supportive evaluation of our manuscript. We appreciate the recognition of the significant collaborative effort involved in coordinating and executing this model intercomparison, as well as the timeliness and relevance of such an exercise in the context of a growing number of sea-ice biogeochemical models. One of our main objectives was indeed to promote transparency and comparability across modelling approaches, and we are encouraged that the reviewer acknowledges the value of running and tuning the models under a common experimental framework.

As the manuscript is currently written, however, it is difficult for a reader outside of the group to gain much insight or understanding from the exercise. I know this is meant to be a "brief communication", but I think it deserves a far more detailed paper, for which the changes would probably be considered major revisions.

We thank the reviewer for their comment. We will try to provide as many details as possible in the revision while staying within the constraints of the Brief Communication format, which we still believe is the most appropriate choice for this exercise because the primary goal is to present key results and insights from the intercomparison in a concise and accessible way, serving as a first step to highlight critical model differences and priorities for future development. A more extensive discussion of model-specific processes and tuning strategies will be better suited to follow-up publications or supplementary material. Please, see also related comments below.

1.) My main concern surrounds the brief treatment of A) describing what the models are doing

We thank the reviewer for their comment. Since all six models are based on previously published and independently developed frameworks, we chose to summarise only their major structural and biogeochemical features in Table 1. However, to enhance clarity and accessibility for readers less familiar with these models, we will include brief descriptions of each model in the Supplementary Material in the revised version of the manuscript.

and B) usefully synthesizing the outcomes. The manuscript sets up its goal as wanting to gain increased process understanding (rather than simply comparing model outputs, which is more the goal of intercomparisons like Watanabe et al. 2019 that they cite) but I did not feel like it went beyond listing individual model tuning in a way that informed my understanding of the sea-ice ecosystem.

We apologise for the lack of clarity regarding our objectives (lines 62-67 and 73-76). Our primary goal in this intercomparison was not to advance new process understanding of the sea-ice ecosystem per se, but rather to improve our understanding of how different models represent key processes and respond to a shared set of boundary conditions. In particular, we aimed to identify where model behaviour diverges, how tuning strategies vary, and what this reveals about current modelling approaches. We will revise the manuscript to better reflect this objective and to clarify how our synthesis highlights model-specific differences and common challenges in simulating sea-ice biogeochemistry.

What were key parameter values before/after tuning? Table 2/the list in L178-182 seems like it should be the heart of the Discussion but I didn't know why modeling teams chose to make those tuning adjustments, whether any choices were specific to model design or previous parameterization, or how model outcomes might be tied to their design. E.g., Can anything be said about models that are BL vs. DE or quota vs. Redfield? Does increased model complexity improve match to observations? Were the tuning changes still within reasonable values for BGC processes, or do they suggest that model physics might be off? What would this work mean for future sea-ice researchers, especially those adjacent to the author group?

We thank the reviewer for this thoughtful and constructive comment. We agree that a deeper synthesis of the tuning outcomes, model structures, and their implications would strengthen the manuscript. In the revised version, we will expand the Discussion to address the following points:

- We will clarify the reasoning behind each model team's parameter adjustments
- We will include a comparative discussion of how different model design choices, such as BL vs. DE formulations, or quota-based vs. Redfield-type stoichiometry, may have influenced both tuning strategies and performance relative to observations.
- We will confirm that the chosen parameter values remained within plausible bounds for sea-ice BGC processes and briefly discuss whether any compensatory tuning may point to mismatches in the physical environment.
- We will enhance the utility and transparency of the intercomparison for the broader community.

Lastly, depending on the direction of Discussion, it may be useful to include more model description than currently exists in the Methods. I understand the challenge of summarizing 6 different models, but sometimes there is a place for including key equations.

We agree that providing additional model details could improve the manuscript's clarity and usefulness to readers. While space constraints limit the extent of methodological detail we can include in the main text, we will add a concise summary of key equations (e.g., primary production, nutrient uptake, and light attenuation) in the Supplementary Material. This will complement the structural descriptions already in Table 1 and the expanded model summaries we are preparing in response to earlier comments.

2.) I would like to see physical variables from both the N-ICE observations and the models (those without prescribed physics). Even though the focus here is on BGC, the ice environment is critical for light and nutrient dynamics and thus for understanding sea-ice algal growth. Please consider adding another plot and adding to the Results and Discussion accordingly. Current places in the writing where the physics were alluded to but could use more backing were L129 and L143-144.

We agree that including physical variables is important for better contextualising the biogeochemical model performance. In the revised manuscript, we will add a figure comparing relevant physical variables from the N-ICE2015 observations with outputs from the models that simulate their own physics. We will also revise the relevant sections of the Results and Discussion to incorporate this comparison and reflect more clearly on how physical variability may have influenced biogeochemical outcomes across models.

3.) It is near impossible to make sense of the nutrient validation when there is only one time point, which the authors themselves acknowledge (L169-172). Have the authors looked into other time series, such as those from Green Edge, CASES (Cape Bathurst), Resolute Bay, etc.?

We agree with the reviewer that a more extended nutrient time series would improve the robustness of model validation. At the time of the experiment, the N-ICE2015 dataset was the only available time series that provided physical and biogeochemical observations at sufficient temporal resolution for all the variables needed to support this intercomparison. However, we acknowledge the value of additional datasets and, also to this end, we plan a second phase of this project.

The writing itself is generally good and clear. Here are a few line-by-line comments:

L8. Tromso is misspelled.

We thank the reviewer for spotting this typo. It will be corrected in the revised manuscript.

L37-38. Please consider adding a citation for the claim of significant effects throughout trophic levels.

We thank the reviewer for this suggestion. In the revised manuscript, we will add citations to support the statement, such as Post et al. (2013), who provide a comprehensive synthesis of

the ecological consequences of Arctic sea-ice decline, including cascading impacts throughout marine food webs:

Post, E., et al. (2013). Ecological consequences of sea-ice decline. Science, 341(6145), 519–524. https://doi.org/10.1126/science.1235225

L51. Please specify that this is maximum algal growth rate.

We thank the reviewer for this suggestion. We will add " algal" to the revised manuscript.

L67. I feel that for the last sentence of the introduction, this places a lot of emphasis on temporal variability, when your results are also about magnitude. Please consider revising.

We thank the reviewer for this helpful observation. In the revised manuscript, we will revise the final sentence of the Introduction, emphasising that the intercomparison explores differences in both the timing and magnitude of simulated sea-ice biogeochemical processes.

L124. Is 83 to 83N correct? If so, please include more details about the drift trajectory

We thank the reviewer for spotting this typo. The correct drift trajectory was from 83°N to 80°N. This will be corrected in the revised manuscript, and we will also include a map of the drift track in the Supplementary Material to provide additional context for the study setup.

L164-166. This sentence is relatively redundant for the information that it conveys. Perhaps trim to "Most models exhibited deviations in either phenology or bloom magnitude."

We thank the reviewer for their suggestion. We will modify the text accordingly in the revised manuscript.

L194-196. This sentence confused me. Something seems off with the "show to disagree" verb?

We thank the reviewer for their comment. We will revise the sentence in the following way: *"Overall, it remains unclear which element primarily limited algal growth, as the models either differ structurally—by excluding the element of interest—or apply different parameterisations."*

L220-221. This statement ("challenges encountering in simulating a refrozen lead") seems important to understanding the models-observations comparison, but it was never discussed before the Conclusion. Please consider treating this in greater detail in the Discussion.

We thank the reviewer for their suggestion. We agree that the unique challenges associated with simulating a refrozen lead warrant more detailed discussion. In the revised manuscript, we will expand the Discussion section to explicitly address these factors and their implications for the observed discrepancies between model outputs and field observations.

L230. This is a minor point, but please consider a more common word than "auspicabile"

We thank the reviewer for their suggestion. We will modify the sentence in the following way in the revised manuscript: *"A Phase 2 of the intercomparison would be highly valuable, potentially extending the study to the habitat variability characteristic of Antarctic sea ice."*

Figure 1. Please report n for the observations and clarify whether replicates are from different ice cores or technical replicates from the same core.

We thank the reviewer for their suggestion. In the revised manuscript, we will report the number of observations (n) for each variable shown in Figure 1 and clarify replicates in the caption.

Table 2. For SIESTA tuning strategy, what does "possibility to keep position" mean?
We thank the reviewer for their comment. It means that algae can actively move against brine movements. We will clarify this in Table 2 of the revised manuscript.

---

## Author Comment (AC2)

**Rebuttal "Brief communication: Intercomparison study reveals pathways for improving the representation of sea-ice biogeochemistry in models" by Tedesco et al., 2025**

Reviewer #2

**General comments**

This study provides an intercomparison of six 1D sea-ice biogeochemical models with a focus on the assessment of simulating spring ice algae blooms and associated nutrient variability. The main findings are that: none of the models adequately captured blooms with their default parameters; tuning improved the ice algae blooms but not the nutrient variability; and more systematic tuning strategies are suggested as a next step. I think it is a great effort to conduct an intercomparison study for sea-ice BGC models, which has not been done except for Watanabe et al. (2019). The manuscript is generally easy to follow and clearly structured. However, it would benefit from more careful proofreading to address minor editorial issues and improve overall readability.

We thank Reviewer #2 for their positive assessment of our work and for recognising the value of this sea-ice biogeochemical model intercomparison. We also appreciate the comment regarding editorial quality. In response, we will carefully proofread the revised manuscript to correct minor editorial issues and improve clarity and readability throughout.

The manuscript type is "brief communication", so I understand that it is written briefly. However, I find it a bit too brief considering the following three points. Therefore, I recommend major revisions and provide suggestions below.

We thank the reviewer for their constructive suggestions. While we chose the Brief Communication format to highlight the core outcomes of the intercomparison in a concise manner, we agree that additional detail would enhance the clarity and utility of the study. In the revised manuscript, we will address the points raised below by expanding key sections, while keeping within the scope and length limitations of the format. Where necessary, we will provide supplementary material to ensure transparency and completeness.

**Physical data. The manuscript lacks the presentation of physical data, even though the text mentions the existence of such data (e.g., L21, L137, L143). Given that physical processes drive the circulation of biogeochemical variables, it seems essential to show the comparison of physical model and observational data, such as snow thickness, ice thickness, and sea surface temperature. With these additions, the study may be able to address (or at least speculate) whether the simulated differences and biases are due to the physical processes.**

We thank the reviewer for their suggestion. We fully agree that including physical data is valuable for interpreting the biogeochemical model performance. In the revised manuscript, we will add a figure comparing observed and simulated physical variables for the models that include their own physical components. We will also expand the Discussion to reflect on how differences in physical conditions may have contributed to the observed model biases and variability.

**Quantitative assessment. Table 2 can be improved by incorporating quantitative findings. Currently, it is a qualitative description that is not very informative and is a bit difficult to follow; one can easily guess the qualitative changes as described in Table 2 (e.g., lower biomass was increased by lowering silica limitation). What would be informative and advance the knowledge is to report the amount of improvements by the amount of parameter adjustments.**

We thank the reviewer for their comment. In the revised manuscript, we will update Table 2 to include specific parameter values before and after tuning (where available), as well as the corresponding changes in key model outputs (e.g., peak chlorophyll-a concentration, bloom timing, nutrient drawdown). We agree that this will allow readers to better assess the magnitude of improvements achieved through tuning and how these relate to parameter adjustments.

**Connection to previous studies. The results and discussion section as well as the conclusions section (L162 onwards) do not appear to contain any reference to previous studies. Hence, it is unclear how this study contributes to the field. This can be achieved by incorporating discussion of the results with previous studies. Specifically, I think that the discussion can be improved by incorporating tuning strategies and intercomparison studies conducted for ocean BGC modelling (e.g., Schartau et al., 2017). Some of these are already mentioned in the manuscript (e.g., L226-234), but it would be better to link these with relevant previous studies to provide a practical direction for future studies.**

Schartau et al. (2017). Reviews and syntheses: parameter identification in marine planktonic ecosystem modelling. Biogeosciences.

We thank the reviewer for their comment. We agree that linking our results more clearly to previous biogeochemical modelling and intercomparison studies would strengthen the context and relevance of our findings. In the revised manuscript, we will expand these parts to explicitly reference relevant works such as Schartau et al. (2017). At the same time, we will emphasise that, to our knowledge, this is the first intercomparison specifically focused on one-dimensional sea-ice biogeochemical models. As such, our study fills a gap in the literature and offers a novel perspective on tuning approaches, model diversity, and shared challenges within the sea-ice modelling community.

**Specific comments**

L21. "N-ICE2015" is too technical for the abstract. It is better to inform the region and season instead (e.g., north of Svalbard during April-June, 2015).

We thank the reviewer for their comment. We will revise the manuscript accordingly.

L22. "tuning" and "adjustments" are used together and they seem to mean the same thing, but this is unclear. I suggest replacing "without tuning, adjustments" by "using their default parameter sets, tuning".

We thank the reviewer for their suggestion. We will revise the manuscript accordingly.

L23. It would be good to add a sentence here to explain why "adjustments improved biomass simulations but had a limited impact on nutrient representation". (at least speculate even though the cause is unknown)

We thank the reviewer for their comment. The limited improvement in nutrient representation compared to biomass is primarily because most model groups prioritised fitting their simulations to the Chl-a observations during the tuning phase, as these data were more temporally resolved and directly linked to the main focus of the study, i.e., the ice algal bloom. In contrast, nutrient observations were limited to a single time point, which made them more difficult to constrain reliably. We will clarify this point in the revised manuscript.

L24. It may be informative to add a few words to describe what "harmonised" means here.

We thank the reviewer for their suggestion. We refer to the development of more coordinated or standardised tuning approaches across models, for example using common performance metrics or agreed-upon parameter bounds. We will revise the sentence to reflect this more explicitly in the following way:

*"Variability in tuning strategies underscores key knowledge gaps and the need for further model development using more coordinated approaches such as common evaluation criteria or shared parameter ranges."*

L28. Should "ice algae" be "bottom ice" instead, given the following phrase "representing the largest biomass fraction in sea ice"?

We thank the reviewer for this comment. However, we respectfully maintain the use of "ice algae" in this sentence. The term refers to the community of microalgae that inhabit the sea ice and is widely used in the literature to describe the biological component responsible for the largest biomass fraction in sea ice (e.g., Poulin et al., 2011). In contrast, "bottom ice" refers to a physical ice layer and not the biological community itself. We will retain the original wording for clarity and consistency with established terminology.

L47. "IAMIP1" should be spelled out.

We thank the reviewer for their comment. In the revised manuscript, we will spell out "IAMIP1" upon first mention as *"Ice Algae Model Intercomparison Project – Phase 1 (IAMIP1)"* to improve clarity for readers unfamiliar with the acronym.

L53. "CMIP6" should be spelled out.

We thank the reviewer for their comment. We will spell out "CMIP6" upon first mention in the revised manuscript as *"Coupled Model Intercomparison Project Phase 6 (CMIP6)"* to ensure clarity for all readers.

L67. I suggest replacing "existing" by "participating", as the former sounds like these are all 1D models that exist.

We thank the reviewer for this suggestion. In the revised manuscript, we will replace "existing" with "participating" to clarify that the six models represent those that were available and contributed to this specific experiment.

L79. "little" or none? Horizontal advection terms are neglected.

We thank the reviewer for this comment. We agree and will include a clarification in the revised manuscript that 1D process models are typically designed to represent vertical processes only, under the assumption that horizontal advection is negligible.

L88. It would be helpful to briefly explain what "dynamic layering" means.

We thank the reviewer for this helpful suggestion. In the revised manuscript, we will briefly explain what is meant by *"dynamic layering"* in this context. Specifically, it refers to the model's ability to adjust the thickness of vertical layers within the sea ice in response to growth and melt processes, thereby allowing for a more realistic representation of habitat structure and biogeochemical gradients.

L92. "Chemical Functional Families (CFF)" does not sound familiar in marine BGC modelling. Please use an alternative term or provide a reference.

We thank the reviewer for their comment. To improve clarity, we will revise the manuscript to use *"Plankton Functional Types (PFTs)"* instead, which more accurately describes the grouping of organisms based on shared functional traits relevant to biogeochemical cycling.

L124. Please correct the latitudinal range "83 to 83 N".

We thank the reviewer for spotting this typo. We will correct the latitudinal range for *"83 to 80°N"* in the revised manuscript.

L126. It is more intuitive to write the range in an increasing order "80.5 and 81.8 N".

We thank the reviewer for their suggestion. To balance clarity and scientific accuracy, we will revise the sentence to explicitly describe the southward drift from 81.8°N to 80.5°N, making both the direction and latitudinal range intuitive for readers: *"Among the four ice floes monitored during the study period, the refrozen lead data were derived from Floe 3, which was studied from mid-April to early June 2015 as it drifted southward from 81.8°N to 80.5°N."*

L139. "Duarte et al. 2017" is one of the participating models in this study? If so, why is the performance poor? Presumably, the model was previously tuned to this study site.

We thank the reviewer for this observation. The model described in Duarte et al. (2017) is indeed one of the participating models in this study. However, it is important to note that the configuration used in our intercomparison did not retain the site-specific tuning applied in the original publication. Instead, all models were initially run using their respective baseline parameterisations, which were designed for broader applicability rather than tailored to the N-ICE2015 lead environment. The poorer performance in the default run reflects this generality and highlights the challenge of transferring model setups across sites without retuning. This underlines the importance of the harmonised tuning phase included in our intercomparison design.

L143. I do not see these physical metrics compared (sea-ice season timing, ice thickness, and snow thickness).

We thank the reviewer for their comment. As already proposed, in the revised manuscript, we will include a comparison of key physical metrics between the N-ICE2015 observations and the models that simulate their own physics. This will be presented in a new figure and discussed in both the Results and Discussion sections to help interpret the influence of physical conditions on biogeochemical model performance.

L149. "the extent of biases" will depend on the location where tuning was conducted for the default parameter sets. Hence, it would be helpful to indicate for which region each model was tuned (in Table 1 and/or the text). This will also give an indication for the "portability" of the model (Sec 2.8 of Friedrichs et al. 2007).

Friedrichs et al. (2007). Assessment of skill and portability in regional marine biogeochemical models: Role of multiple planktonic groups, JGR-Oceans.

We thank the reviewer for their suggestion. We agree that the extent of model biases in the default runs can be influenced by the region for which each model was originally tuned, and that this information is important for evaluating model portability. In the revised manuscript, we will add a column to Table 1 indicating the geographic region or study site associated with the original tuning of each model's default parameter set.

L158. Please describe the source of the atmospheric forcing used here.

We thank the reviewer for their comment. In the revised manuscript, we will add a description of the atmospheric forcing used in the experiment. Specifically, the models were forced with atmospheric data collected directly during the N-ICE expedition. This information will be included in the Methods section for clarity and completeness.

L159. Please specify which model is the one without a thermodynamic component.

We thank the reviewer for their suggestion. In the revised manuscript, we will specify that the model without a thermodynamic component is SIMBA. Unlike the other models,

it relies on prescribed physical fields rather than simulating them dynamically. This distinction will be clearly stated in the Methods section and reflected in Table 1.

L180. "Change in the initial simulation date" seems strange to be considered a tuning parameter.

We thank the reviewer for this comment. We agree that a change in the initial simulation date is not a tuning parameter in the conventional sense. We will revise the text to clarify that this change was a modelling decision made to improve alignment with observational context, rather than a formal tuning of model parameters.

---

## Author Response (AR1)

**Rebuttal** "Brief communication: Intercomparison study reveals pathways for improving the representation of sea-ice biogeochemistry in models" by Tedesco et al., 2025

The manuscript presents an intercomparison between 6 one-dimensional sea-ice biogeochemical models. Models were forced with the same atmospheric and oceanic data from a time series of observations from a refrozen lead and validated with associated chlorophyll *a*, nitrate, and silicate data. Models were first run with their default parameterizations, and then teams were asked to tune their models to best match the observations. The authors found that with tuning, most models were able to reproduce chlorophyll *a* timing and concentration but not so much nutrients. Model teams also differed in their tuning choices.

I think the exercise of model intercomparison is valuable and this is a timely moment to do one for sea-ice BGC, as the number of models has been increasing. I also think the exercise is well-structured and shows significant effort on the part of many people in independently running and tuning the models with the same set of forcings (not a given in other model intercomparisons) in order to reveal a process understanding.

We thank Reviewer #1 for their thoughtful and supportive evaluation of our manuscript. We appreciate the recognition of the significant collaborative effort involved in coordinating and executing this model intercomparison, as well as the timeliness and relevance of such an exercise in the context of a growing number of sea-ice biogeochemical models. One of our main objectives is indeed to promote transparency and comparability across modelling approaches, and we are encouraged that the reviewer acknowledges the value of running and tuning the models under a common experimental framework.

As the manuscript is currently written, however, it is difficult for a reader outside of the group to gain much insight or understanding from the exercise. I know this is meant to be a "brief communication", but I think it deserves a far more detailed paper, for which the changes would probably be considered major revisions.

We thank the reviewer for their comment. We have revised the manuscript, adding more details while staying within the constraints of the Brief Communication format, which we still believe is the most appropriate choice for this exercise because the primary goal is to present key results and insights from the intercomparison in a concise and accessible way, serving as a first step to highlight critical model differences and priorities for future development. A more extensive discussion of model-specific processes and tuning strategies is provided in the new Supplementary Material. Please see also the related comments below.

1.) My main concern surrounds the brief treatment of A) describing what the models are doing

We thank the reviewer for their comment. Since all six models are based on previously published and independently developed frameworks, we chose to summarise only their major features in Table 1. However, to enhance clarity and accessibility for readers less familiar with these models, we've included a description of each model in the Supplementary Material in the revised version of the manuscript.

and B) usefully synthesizing the outcomes. The manuscript sets up its goal as wanting to gain increased process understanding (rather than simply comparing model outputs, which is more the goal of intercomparisons like Watanabe et al. 2019 that they cite) but I did not feel like it went beyond listing individual model tuning in a way that informed my understanding of the sea-ice ecosystem.

We apologise if our objectives were not clear earlier. Our primary goal in this intercomparison was not to advance new process understanding of the sea-ice ecosystem per se, but rather to improve our understanding of how different models represent key processes and respond to a shared set of boundary conditions. In particular, we aimed to identify where model behaviour diverges, how tuning strategies vary, and what this reveals about current modelling approaches. We have revised the manuscript to reflect this objective better and to clarify how our synthesis highlights model-specific differences and common challenges in simulating sea-ice biogeochemistry. Please, see lines 65-70 of the tracked changes revised ms.

What were key parameter values before/after tuning? Table 2/the list in L178-182 seems like it should be the heart of the Discussion but I didn't know why modeling teams chose to make those tuning adjustments, whether any choices were specific to model design or previous parameterization, or how model outcomes might be tied to their design. E.g., Can anything be said about models that are BL vs. DE or quota vs. Redfield? Does increased model complexity improve match to observations? Were the tuning changes still within reasonable values for BGC processes, or do they suggest that model physics might be off? What would this work mean for future sea-ice researchers, especially those adjacent to the author group?

We thank the reviewer for this thoughtful and constructive comment. We agree that a deeper synthesis of the tuning outcomes, model structures, and their implications can strengthen the manuscript. In the revised version, we have expanded the Discussion to address the following points:

- We have clarified the reasoning for parameter adjustments. Please, see lines 230-243 and lines 256-280 of the tracked changes revised ms.
- We have showed the chosen parameter and their values before and after tuning in Table 2.

Lastly, depending on the direction of Discussion, it may be useful to include more model description than currently exists in the Methods. I understand the challenge of summarizing 6 different models, but sometimes there is a place for including key equations.

We agree that providing additional model details can improve the manuscript's clarity and usefulness to readers. While space constraints limit the extent of methodological detail we

can include in the main text, we have added an essential summary of the models in the Supplementary Material. This complements the structural descriptions in Table 1, now expanded (lines 110-118 of the tracked changes revised ms).

2.) I would like to see physical variables from both the N-ICE observations and the models (those without prescribed physics). Even though the focus here is on BGC, the ice environment is critical for light and nutrient dynamics and thus for understanding sea-ice algal growth. Please consider adding another plot and adding to the Results and Discussion accordingly. Current places in the writing where the physics were alluded to but could use more backing were L129 and L143-144.

We agree that including physical variables is important for better contextualising the biogeochemical model performance. In the revised manuscript, we have added a figure (Fig. 1) comparing relevant physical variables (ice thickness and air temperature) from the N-ICE2015 observations with outputs from the models that simulate their own physics. We have also revised the relevant sections of the Results and Discussion to incorporate this comparison and reflect more clearly on how physical variability could have influenced biogeochemical outcomes across models (lines 183-195, lines 207-209 of the tracked changes revised ms). This was also a request from Reviewer #2.

3.) It is near impossible to make sense of the nutrient validation when there is only one time point, which the authors themselves acknowledge (L169-172). Have the authors looked into other time series, such as those from Green Edge, CASES (Cape Bathurst), Resolute Bay, etc.?

We agree with the reviewer that a more extended nutrient time series would improve the robustness of model validation. During the N-ICE expedition additional samples from the refrozen lead were taken for nutrient analysis. However, those ice cores were much thicker than those where sea-ice Chl-a was measured, so we agreed to exclude them for consistency but focused on order of magnitude of concentration of nutrients where data were available.

Furthermore, at the time of the experiment, the N-ICE2015 dataset was the only available time series that provided physical and biogeochemical observations at sufficient temporal resolution for all the variables needed to support this intercomparison, including initial and boundary conditions.

However, we acknowledge the value of additional datasets and, also to this end, we plan a second phase of this project. We have further addressed this aspect in the revised manuscript at lines 308-312 of the tracked changes revised ms.

The writing itself is generally good and clear. Here are a few line-by-line comments:

L8. Tromso is misspelled.

We thank the reviewer for spotting this typo. It is now corrected (line 8).

L37-38. Please consider adding a citation for the claim of significant effects throughout trophic levels.

We thank the reviewer for this suggestion. In the revised manuscript, we have added citations to support the statement, such as Post et al. (2013) and Kotch et al. (2023), who provide a comprehensive synthesis of the ecological consequences of sea-ice decline, including cascading impacts throughout marine food webs:

Post, E., et al. (2013). Ecological consequences of sea-ice decline. Science, 341(6145), 519–524. https://doi.org/10.1126/science.1235225.

Koch, C.W., Brown, T.A., Amiraux, R. *et al.* Year-round utilization of sea ice-associated carbon in Arctic ecosystems. *Nat Commun* 14, 1964 (2023). https://doi.org/10.1038/s41467-023-37612-8

L51. Please specify that this is maximum algal growth rate.

We thank the reviewer for this suggestion. We've added " algal" to the revised manuscript.

L67. I feel that for the last sentence of the introduction, this places a lot of emphasis on temporal variability, when your results are also about magnitude. Please consider revising.

We thank the reviewer for this helpful observation. In the revised manuscript, we've revised the final sentence of the Introduction, adding "and magnitude" to the sentence.

L124. Is 83 to 83N correct? If so, please include more details about the drift trajectory

We thank the reviewer for spotting this typo. The correct drift trajectory was from 83°N to 80°N. This has been corrected in the revised manuscript, and we have also included a map of the drift track in the Supplementary Material to provide additional context for the study setup (Figure S1).

L164-166. This sentence is relatively redundant for the information that it conveys. Perhaps trim to "Most models exhibited deviations in either phenology or bloom magnitude."

We thank the reviewer for their suggestion. We've modified the text in the revised manuscript (lines 203-205 of the tracked changes revised ms).

L194-196. This sentence confused me. Something seems off with the "show to disagree" verb?

We thank the reviewer for their comment. We've revised this part and extended for clarity. Please, see lines 238-243 of the tracked changes revised ms.

L220-221. This statement ("challenges encountering in simulating a refrozen lead") seems important to understanding the models-observations comparison, but it was never discussed before the Conclusion. Please consider treating this in greater detail in the Discussion.

We thank the reviewer for their suggestion. We agree that the unique challenges associated with simulating a refrozen lead warrant more detailed discussion. In the revised manuscript, we've expanded the Discussion section to explicitly address these factors and their implications for the observed discrepancies between model outputs and field observations (lines 262-267 of the tracked changes revised ms).

L230. This is a minor point, but please consider a more common word than "auspicabile"

We thank the reviewer for their suggestion. We've modified the sentence in the following way in the revised manuscript: *"A Phase 2 of the intercomparison would be highly valuable..."*

Figure 1. Please report n for the observations and clarify whether replicates are from different ice cores or technical replicates from the same core.

We thank the reviewer for their suggestion. In the revised manuscript, we've clarified in the caption the number of observations (at least 5 esch) and that replicates are from different ice cores.

Table 2. For SIESTA tuning strategy, what does "possibility to keep position" mean?
We thank the reviewer for their comment. It means that algae can actively move against brine movements. We've clarified this in Table 2 of the revised manuscript.

Reviewer #2

**General comments**

This study provides an intercomparison of six 1D sea-ice biogeochemical models with a focus on the assessment of simulating spring ice algae blooms and associated nutrient variability. The main findings are that: none of the models adequately captured blooms with their default parameters; tuning improved the ice algae blooms but not the nutrient variability; and more systematic tuning strategies are suggested as a next step. I think it is a great effort to conduct an intercomparison study for sea-ice BGC models, which has not been done except for Watanabe et al. (2019). The manuscript is generally easy to follow and clearly structured. However, it would benefit from more careful proofreading to address minor editorial issues and improve overall readability.

We thank Reviewer #2 for their positive assessment of our work and for recognising the value of this sea-ice biogeochemical model intercomparison. We also appreciate the comment regarding editorial quality. In response, we've carefully proofread the revised manuscript.

The manuscript type is "brief communication", so I understand that it is written briefly. However, I find it a bit too brief considering the following three points. Therefore, I recommend major revisions and provide suggestions below.

We thank the reviewer for their constructive suggestions. While we chose the Brief Communication format to highlight the core outcomes of the intercomparison in a concise manner, we agree that additional detail would enhance the clarity and utility of the study. In the revised manuscript, we've addressed the points raised below by expanding key sections, while keeping within the scope and length limitations of the format. Where necessary, we've provided supplementary material to ensure transparency and completeness.

**Physical data. The manuscript lacks the presentation of physical data, even though the text mentions the existence of such data (e.g., L21, L137, L143). Given that physical processes drive the circulation of biogeochemical variables, it seems essential to show the comparison of physical model and observational data, such as snow thickness, ice thickness, and sea surface temperature. With these additions, the study may be able to address (or at least speculate) whether the simulated differences and biases are due to the physical processes.**

We thank the reviewer for their suggestion. We fully agree that including physical data is valuable for interpreting the performance of the biogeochemical model. In the revised manuscript, we've added a figure (Fig. 1) comparing observed and simulated physical variables for the models that include their own physical components, in particular sea-ice thickness and surface temperature. We did not include snow depth as it was little (2-6 cm) and little variable. We've also expanded the Results and Discussion section to reflect on how differences in physical conditions have contributed to the observed model

biases and variability (lines 183-195, 207-209, 262-267 of the tracked changes revised ms).

**Quantitative assessment. Table 2 can be improved by incorporating quantitative findings. Currently, it is a qualitative description that is not very informative and is a bit difficult to follow; one can easily guess the qualitative changes as described in Table 2 (e.g., lower biomass was increased by lowering silica limitation). What would be informative and advance the knowledge is to report the amount of improvements by the amount of parameter adjustments.**

We thank the reviewer for their comment. In the revised manuscript, we've updated Table 2 to include specific parameter values before and after tuning, as well as the corresponding changes in key model outputs (e.g., peak chlorophyll-a concentration, peak bloom timing). We agree that this will allow readers to better assess the magnitude of improvements achieved through tuning and how these relate to parameter adjustments.

**Connection to previous studies. The results and discussion section as well as the conclusions section (L162 onwards) do not appear to contain any reference to previous studies. Hence, it is unclear how this study contributes to the field. This can be achieved by incorporating discussion of the results with previous studies. Specifically, I think that the discussion can be improved by incorporating tuning strategies and intercomparison studies conducted for ocean BGC modelling (e.g., Schartau et al., 2017). Some of these are already mentioned in the manuscript (e.g., L226-234), but it would be better to link these with relevant previous studies to provide a practical direction for future studies.**

Schartau et al. (2017). Reviews and syntheses: parameter identification in marine planktonic ecosystem modelling. Biogeosciences.

We thank the reviewer for their comment and this new reference. We agree that linking our results more clearly to previous biogeochemical modelling and intercomparison studies would strengthen the context and relevance of our findings. In the revised manuscript, we've emphasised that this is the first intercomparison specifically focused on one-dimensional sea-ice biogeochemical models. As such, our study fills a gap in the literature and offers a novel perspective on tuning approaches, model diversity, and shared challenges within the sea-ice and marine biogeochemical modelling community. Please see lines (308-311) of the tracked changes revised version of the ms.

**Specific comments**

L21. "N-ICE2015" is too technical for the abstract. It is better to inform the region and season instead (e.g., north of Svalbard during April-June, 2015).

We thank the reviewer for their comment. We've revised the manuscript with "Arctic drift expedition in spring 2015".

L22. "tuning" and "adjustments" are used together and they seem to mean the same thing, but this is unclear. I suggest replacing "without tuning, adjustments" by "using their default parameter sets, tuning".

We thank the reviewer for their suggestion. We've revised the manuscript accordingly.

L23. It would be good to add a sentence here to explain why "adjustments improved biomass simulations but had a limited impact on nutrient representation". (at least speculate even though the cause is unknown)

We thank the reviewer for their comment. The limited improvement in nutrient representation compared to biomass is primarily because most model groups prioritised fitting their simulations to the Chl-a observations during the tuning phase, as these data were more temporally resolved and directly linked to the main focus of the study, i.e., the ice algal bloom. In contrast, nutrient observations were limited to a single time point, which made them more difficult to constrain reliably. We are limited in the number of words we can use for the abstract, but we've made sure to clarify this point in the Results and Discussion section of the revised manuscript (lines 245-250 of the tracked changes revised ms).

L24. It may be informative to add a few words to describe what "harmonised" means here.

We thank the reviewer for their suggestion. We refer to the development of more coordinated or standardised tuning approaches across models, for example using common performance metrics or agreed-upon parameter bounds. We've revised the manuscript (lines 308-313 of the tracked changes revised ms), adding a sentence in the Conclusion section to reflect this more explicitly in the following way.

L28. Should "ice algae" be "bottom ice" instead, given the following phrase "representing the largest biomass fraction in sea ice"?

We thank the reviewer for this comment. However, we respectfully maintain the use of "ice algae" in this sentence. The term refers to the community of microalgae that inhabit the sea ice and is widely used in the literature to describe the biological component responsible for the largest biomass fraction in sea ice (e.g., Poulin et al., 2011). In contrast, "bottom ice" refers to a physical ice layer and not the biological community itself. We will retain the original wording for clarity and consistency with established terminology.

Poulin, M., Daugbjerg, N., Gradinger, R., Ilyash, L. V., Ratkova, T. N., von Quillfeldt, C.: The pan-Arctic biodiversity of marine pelagic and sea-ice unicellular eukaryotes: a first-attempt assessment, Mar. Biodiv., 41, 13–28, doi:10.1007/s12526-010-0058-8, 2011.

L47. "IAMIP1" should be spelled out.

We thank the reviewer for their comment. In the revised manuscript, we've spelled out "IAMIP1" upon first mention as *"Ice Algae Model Intercomparison Project – Phase 1 (IAMIP1)"* to improve clarity for readers unfamiliar with the acronym.

L53. "CMIP6" should be spelled out.

We thank the reviewer for their comment. We've spelled out "CMIP6" upon first mention in the revised manuscript as *"CMIP6 (Coupled Model Intercomparison Project Phase 6, Eyring et al., 2016)"* to ensure clarity for all readers.

Eyring, V., Bony, S., Meehl, G. A., Senior, C. A., Stevens, B., Stouffer, R. J., and Taylor, K. E.: Overview of the Coupled Model Intercomparison Project Phase 6 (CMIP6) experimental design and organization, Geosci. Model Dev., 9, 1937-1958, doi:10.5194/gmd-9-1937-2016, 2016.

L67. I suggest replacing "existing" by "participating", as the former sounds like these are all 1D models that exist.

We thank the reviewer for this suggestion. In the revised manuscript, we've replaced "existing" with "participating" to clarify that the six models represent those that were available and contributed to this specific experiment.

L79. "little" or none? Horizontal advection terms are neglected.

We thank the reviewer for this comment. We agree and have included a clarification in the revised manuscript that 1D process models are typically designed to represent vertical processes only, under the assumption that horizontal advection is negligible.

L88. It would be helpful to briefly explain what "dynamic layering" means.

We thank the reviewer for this helpful suggestion. In the revised manuscript, we've briefly explained what is meant by *"dynamic layering"* in this context (lines 93-95 of the tracked changes revised ms). Specifically, it refers to the model's ability to adjust the thickness of vertical layers within the sea ice in response to growth and melt processes, thereby allowing for a more realistic representation of habitat structure and biogeochemical dynamics.

L92. "Chemical Functional Families (CFF)" does not sound familiar in marine BGC modelling. Please use an alternative term or provide a reference.

We thank the reviewer for their comment. To improve clarity, we've revised the manuscript to use *"Plankton Functional Types (PFTs)"* instead, which is more commonly used to describe the grouping of organisms based on shared functional traits relevant to biogeochemical cycling.

L124. Please correct the latitudinal range "83 to 83 N".

We thank the reviewer for spotting this typo. We've corrected the latitudinal range for *"83 to 80°N"* in the revised manuscript.

L126. It is more intuitive to write the range in an increasing order "80.5 and 81.8 N".

We thank the reviewer for their suggestion. To balance clarity and scientific accuracy, we've revised the sentence to explicitly describe the southward drift from 81.8°N to 80.5°N, making both the direction and latitudinal range intuitive for readers: *"Among the four ice floes monitored during the study period, the refrozen lead data were derived from Floe 3, which was studied from mid-April to early June 2015 as it drifted southward from 81.8°N to 80.5°N."*

L139. "Duarte et al. 2017" is one of the participating models in this study? If so, why is the performance poor? Presumably, the model was previously tuned to this study site.

We thank the reviewer for this observation. The model described in Duarte et al. (2017) is indeed one of the participating models in this study. However, it is important to note that the configuration used in our intercomparison did not retain the site-specific tuning applied in the original publication. Instead, all models were initially run using their respective baseline parameterisations, which were designed for broader applicability rather than tailored to the N-ICE2015 lead environment. The poorer performance in the default run reflects this generality and highlights the challenge of transferring model setups across sites without retuning. This underlines the importance of the harmonised tuning phase included in our intercomparison design. To make this aspect clearer, we have revised this sentence and replaced "*successfully used*" with "*tested for feasibility*".

L143. I do not see these physical metrics compared (sea-ice season timing, ice thickness, and snow thickness).

We thank the reviewer for their comment. As already proposed, in the revised manuscript, we've included a comparison of key physical metrics between the N-ICE2015 observations and the models that simulate their own physics. This is now presented in a new figure (Fig. 1) and in the Results and Discussion section to help interpret the influence of physical conditions on biogeochemical model performance.

L149. "the extent of biases" will depend on the location where tuning was conducted for the default parameter sets. Hence, it would be helpful to indicate for which region each model was tuned (in Table 1 and/or the text). This will also give an indication for the "portability" of the model (Sec 2.8 of Friedrichs et al. 2007).

Friedrichs et al. (2007). Assessment of skill and portability in regional marine biogeochemical models: Role of multiple planktonic groups, JGR-Oceans.

We thank the reviewer for their suggestion. We agree that the extent of model biases in the default runs can be influenced by the region for which each model was originally tuned, and that this information is important for evaluating model portability. In the

revised manuscript, we've added a column to Table 1 indicating the geographic region or study site associated with the original tuning of each model's default parameter set.

L158. Please describe the source of the atmospheric forcing used here.

We thank the reviewer for their comment. In the revised manuscript, we've added a description of the atmospheric forcing used in the experiment. Specifically, the models were forced with atmospheric data collected directly during the N-ICE expedition. This information is now included in the Methods section (lines 173-178 of the tracked changes revised ms).

L159. Please specify which model is the one without a thermodynamic component.

We thank the reviewer for their suggestion. In the revised manuscript, we've specified that the model without a thermodynamic component is SIMBA. Unlike the other models, it relies on prescribed physical fields rather than simulating them dynamically. This distinction is now clearly stated in the Methods section and reflected in Table 1.

L180. "Change in the initial simulation date" seems strange to be considered a tuning parameter.

We thank the reviewer for this comment. We've removed this form the text because it was not applied in the end.